# The *Staphylococcus aureus* cell division protein, DivIC, interacts with the cell wall and controls its biosynthesis

Mariana Tinajero-Trejo[1,2,5], Oliver Carnell[1,2,5], Azhar F. Kabli[1,2], Laia Pasquina-Lemonche[2,3], Lucia Lafage [1,2], Aidong Han [4], Jamie K. Hobbs [2,3] & Simon J. Foster [1,2✉]

Bacterial cell division is a complex, dynamic process that requires multiple protein components to orchestrate its progression. Many division proteins are highly conserved across bacterial species alluding to a common, basic mechanism. Central to division is a transmembrane trimeric complex involving DivIB, DivIC and FtsL in Gram-positives. Here, we show a distinct, essential role for DivIC in division and survival of *Staphylococcus aureus*. DivIC spatially regulates peptidoglycan synthesis, and consequently cell wall architecture, by influencing the recruitment to the division septum of the major peptidoglycan synthetases PBP2 and FtsW. Both the function of DivIC and its recruitment to the division site depend on its extracellular domain, which interacts with the cell wall via binding to wall teichoic acids. DivIC facilitates the spatial and temporal coordination of peptidoglycan synthesis with the developing architecture of the septum during cell division. A better understanding of the cell division mechanisms in *S. aureus* and other pathogenic microorganisms can provide possibilities for the development of new, more effective treatments for bacterial infections.

[1] School of Biosciences, University of Sheffield, Sheffield, UK. [2] The Florey Institute for Host-Pathogen Interactions, University of Sheffield, Sheffield, UK. [3] Department of Physics and Astronomy, University of Sheffield, Sheffield, UK. [4] State Key Laboratory of Cellular Stress Biology, School of Life Sciences, Xiamen University, Xiamen, China. [5] These authors contributed equally: Mariana Tinajero-Trejo, Oliver Carnell. ✉email: s.foster@sheffield.ac.uk

Cell division is a fundamental facet of the bacterial lifecycle whereby the synthesis of a cell wall septum, prior to scission, results in 2 daughter cells. This overall simplicity necessitates a complex ensemble of protein players required for the synthesis of cell wall components, acting not only at the appropriate place, but also time, to ensure accurate septation and maintenance of cellular integrity. We are beginning to understand the roles of many of the biosynthetic enzymes involved in cell division[1,2], but little is known as to how they are orchestrated temporally and spatially. In *S. aureus*, the cell wall is mainly composed of a network of highly crosslinked layers of peptidoglycan (PG)[3], containing large amounts of phosphate-rich glycopolymers either attached to the cytoplasmic membrane (lipoteichoic acids, LTA) or covalently bound to PG (wall teichoic acids, WTA)[4,5]. Cell division requires a finely tuned regulation of cell wall homoeostasis to maintain structural integrity during morphogenesis[6]. Those protein components that are collectively involved in division are called the divisome[7], where the high level of protein conservation across bacteria suggests a common, underlying division mechanism[2,8].

*S. aureus* has been a useful subject for the study of cell structure and division mechanisms for several years[2], where the lack of cylindrical elongation of the cell wall makes it an apparent simple system[9,10]. Cell division is characterised by cell wall morphological checkpoints that mark its successful progression[9,11]. Firstly, a thick band of PG is synthesised at the division site called the piecrust[9]. This is followed by transition to septal plate formation. The developing septum is initially "V" shaped, that is filled out to give parallel sides after closing of the septal annulus[12]. Only then do the daughter cells separate as a result of a rapid scission[13], revealing a PG architecture of tightly packed concentric rings on the newly exposed septal surface but a fine disordered mesh on the membrane-facing side[11]. The complex coordination of PG architectures requires multiple components, but *S. aureus* has only two essential penicillin binding proteins (PBP1 and PBP2) that are required for the final, transpeptidation stage of PG biosynthesis and hence cell division[14,15]. The activities of these PBPs are thus highly coordinated as their inhibition during division, via the action of β-lactam antibiotics, results in loss of control of PG homoeostasis, catastrophic failure of the cell wall structure and death[16]. Thus, spatial and temporal control of cell division are an absolute requirement for its successful progression, requiring an intimate relationship between cell wall morphogenesis and the divisome. In *S. aureus*, this is supported by the molecular localisation of key divisome components across the developing septum to allow the efficient completion of such a complex feat of bioengineering[12].

The arrival of the tubulin homologue FtsZ, the most highly conserved protein in the divisome, to the cell division plane sets in motion the recruitment and assembly of essential and auxiliary proteins during early cell division[17]. At a later stage, the membrane-associated proteins DivIC, DivIB and FtsL are recruited[17,18]. With the exception of a few obligate intracellular species[19], and the wall-less bacterium *Mycoplasma*[20], DivIB, DivIC and FtsL are conserved across bacteria[21–23]. This, together with the formation of a transmembrane trimeric complex in Gram-positives, and their orthologues (FtsQ, FtsB and FtsL) in Gram-negatives[21–23] strongly suggest a functional importance during division. We have previously shown that DivIB, a PG-binding protein, is essential for septal completion in *S. aureus*[24]; however how it, and its cell division partners, control morphogenesis remains mostly unknown. Here we have established DivIC as a key mediator of cell wall dynamics during the division process. Loss of DivIC results in reduced septal PG synthesis and death of the cells.

## Results

### *S. aureus* DivIC is essential for growth, viability and cell size control.

A high-density transposon screen identified *S. aureus* *divIC* (SAOUHSC_00482) as putatively essential[25]. To test this, a conditional lethal strain was constructed carrying an extra copy of *divIC* under control of the IPTG inducible promoter ($P_{spac}$) inserted at the lipase locus (*geh*::$P_{spac}$-*divIC*) and replaced the native *divIC* with a tetracycline resistance cassette (henceforth referred to as Δ*divIC*) (Fig. 1a). A western blot confirmed the expression of DivIC in cultures grown in the presence but not the absence of IPTG (Fig. 1b). Growth defects in cells depleted of DivIC were investigated by comparing cultures of Δ*divIC* incubated with and without IPTG. A previously characterised, IPTG-dependent *divIB* conditional lethal (Δ*divIB*)[24] and the isogenic strain (SH1000) were used for comparison. In the absence of IPTG, Δ*divIC* had a drastically reduced plating efficiency on agar (Supplementary Fig. 1), and in liquid media growth stopped (Fig. 1c top) and the cells lost viability (Fig.1c, bottom) after 3 h culture. Growth of Δ*divIC* cultures containing IPTG was comparable to that of the isogenic strain, supporting the essentiality of DivIC for cell growth and viability (Fig. 1c). The Δ*divIC* strain grown with IPTG may not have a phenotype identical to SH1000 (the parental strain) as the $P_{spac}$ promoter will lead to *divIC* expression levels likely different to WT.

Fluorescence microscopy was used to examine the effect of loss of DivIC on cell morphology. Cells of the Δ*divIC* strain grown for 2 and 3 h in the absence of IPTG showed significantly larger volumes (~1.8 and ~3.2-fold respectively) compared to SH1000 ($P < 0.0001$) (Fig. 1e). The increase in volume of the DivIB-depleted cells[24] was however much more pronounced (~6 fold greater compared to SH1000 in 2 h cultures) (Fig. 1d, e). Bulging, multi-septation and lysis, not observed in DivIC-lacking cells, prevented determination of cell volumes at 3 h for Δ*divIB* (Fig.1d, e). These results suggest that the functions of DivIC and DivIB in cell division may be linked but are not entirely co-dependent in *S. aureus*.

### DivIC and DivIB are not required for divisome protein stability.

There is a strong interdependence among DivIC, DivIB and FtsL to promote stabilisation of the trimeric complex and for their localisation and recruitment of other proteins to the division site in *B. subtilis* and *E. coli*[23,26–33]. Thus, whether the essential role of DivIC could result from premature degradation of divisome proteins was explored. Using immunoblots to compare the levels of DivIC, DivIB, FtsL, FtsZ, PBP1 and PBP2 in lysates of Δ*divIC* grown with or without IPTG, no decrease in the overall protein levels between lysates from SH1000 and Δ*divIC* cultures without IPTG was found. Similarly, there was no obvious protein degradation when lysates from Δ*divIB* were tested. Interestingly, after 3 h, significantly higher levels of FtsL were detected in lysates of cultures lacking either DivIC or DivIB suggesting a role for these proteins in maintaining low levels of FtsL (Supplementary Fig. 2).

### Loss of DivIC affects cell division and wall architecture.

The involvement of DivIC in cell division was examined by comparing cells grown with and without IPTG for 2 and 3 h. The fluorescent D-amino acid derivative HADA was used to label PG biosynthesis[34] (Fig. 2a). Depletion of DivIC led to an increase in cells with incomplete septa (~10% of total) (Fig. 2b). Interestingly, only after 3 h in the absence of IPTG, DivIC-depleted cells showed an increase in septal defects (to ~15% of the total cells compared to the plus IPTG control) (Fig. 2b), mainly consisting of aberrant septation (Fig. 2a). Further cell division analysis was performed by sequential labelling of PG synthesis using two distinct fluorescent D-amino acid derivatives[12,34,35]. Cell cycle progression was determined by first labelling with HADA (5 min

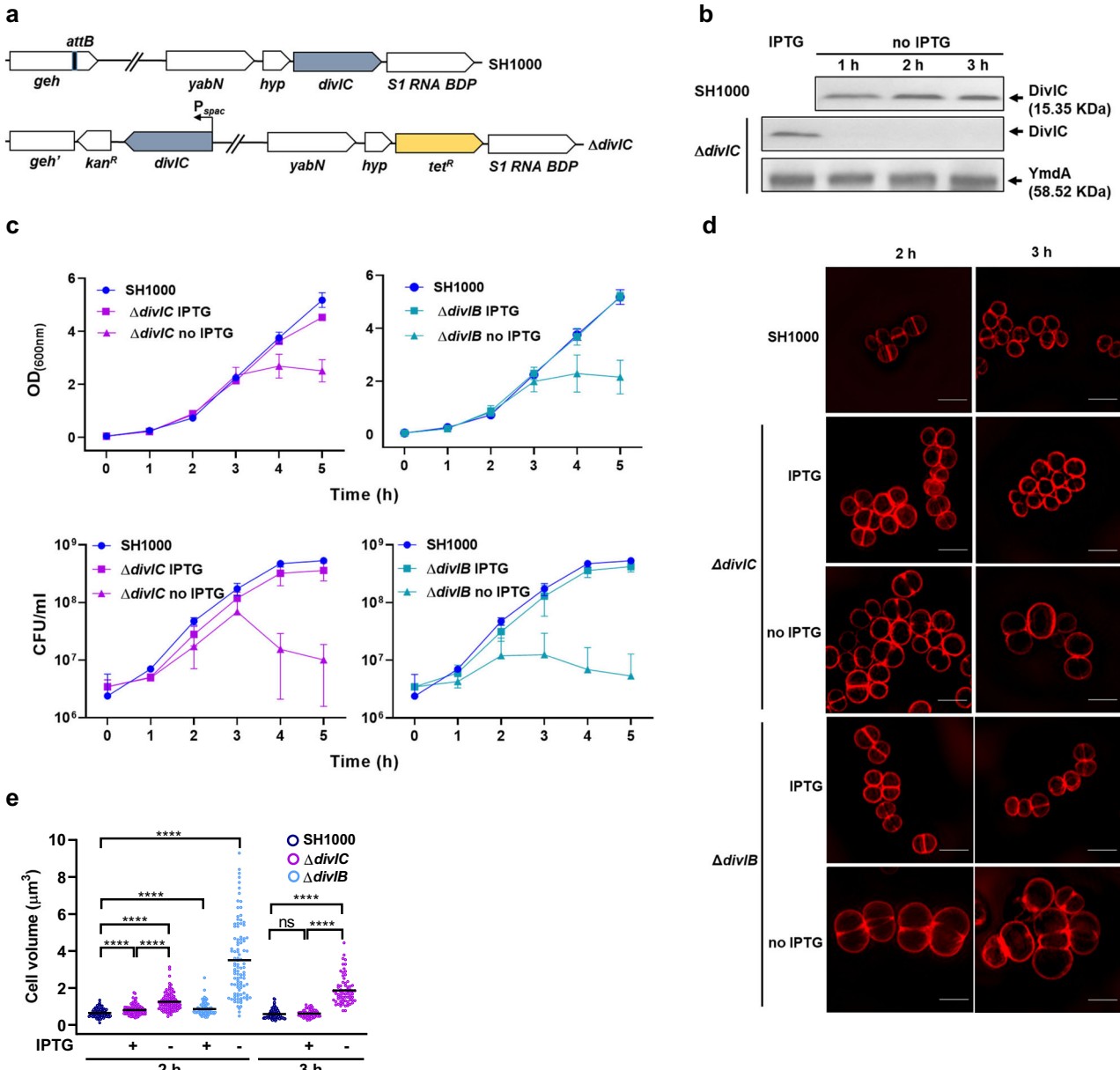

**Fig. 1 DivIC is essential for growth, viability and cell size regulation. a** Genetic map of the parental (SH1000) and the *divIC* conditional lethal (Δ*divIC*, SJF5450) strains. **b** Western blot showing levels of DivIC (15.35 KDa) in SH1000, Δ*divIC* in the presence of IPTG or after 1, 2 and 3 h incubation in the absence of IPTG. Anti-DivIC antibodies were used for detection. Levels of YmdA were detected using anti-YmdA as loading control. **c** Cell growth (top) and viability (bottom) of SH1000, Δ*divIC* and Δ*divIB* (SJF3883) strains in the presence and absence of IPTG. Graphs show mean and standard deviations of three independent experiments. **d** Structured Illumination Microscopy (SIM) images of SH1000, Δ*divIC* and Δ*divIB* cells grown for 2 and 3 h with and without IPTG and treated with NHS-ester 555 to stain cell walls. Images are average intensity projections of z stacks. Scale bars, 2 µm. Data is representative of two independent experiments. **e** Cell volumes of SH1000, Δ*divIC* and Δ*divIB* cells grown with and without IPTG for 2 and 3 h (SH1000 and Δ*divIC* only), determined from SIM images of cells stained with NHS-ester 555. Each circle indicates a single cell. $n = 100$ cells for samples taken a 2 h and SH1000 at 3 h, $n = 62$ and 63 for 3 h plus and minus IPTG respectively. *P*-values were determined by Mann–Whitney *U*-tests (****$P < 0.0001$).

pulse), followed by incubation in label-free medium (15 min) and finally with the dipeptide ADA-DA (5 min chase) (Fig. 2c). HADA allowed the identification of cells with no septum, an incomplete septum, or a complete septum, while ADA-DA signal indicated the progression into subsequent stages of the cell cycle (Fig. 2d). In the absence of DivIC, ~10% of the cells identified by HADA labelling as originally with no septum failed to progress into the next stages of cell division (compared to only ~1% in cells expressing the protein). An accumulation of cells with an incomplete septum (~14% more of the total compared to control cells) was also observed. Only a small proportion of cells, which

in cultures grown with IPTG accounted for ~30% of the total, reached the stage of septal completion (~4%) or showed signs of cell splitting (~1%), (Fig. 2d left). Next, the role of DivIC in the progression of the population of cells identified as with an incomplete septum (by HADA labelling) was analysed (Fig. 2d centre). Only ~37% of the total, of the DivIC depleted cells, had reached the splitting stage after septation (by ADA-DA labelling) compared to ~84% for SH1000 (Fig. 2d centre), indicating a delay in septal development. The Δ*divIC* cells grown with IPTG also showed a delay to transit from incomplete septum to separation (~54%) (Fig. 2d centre).

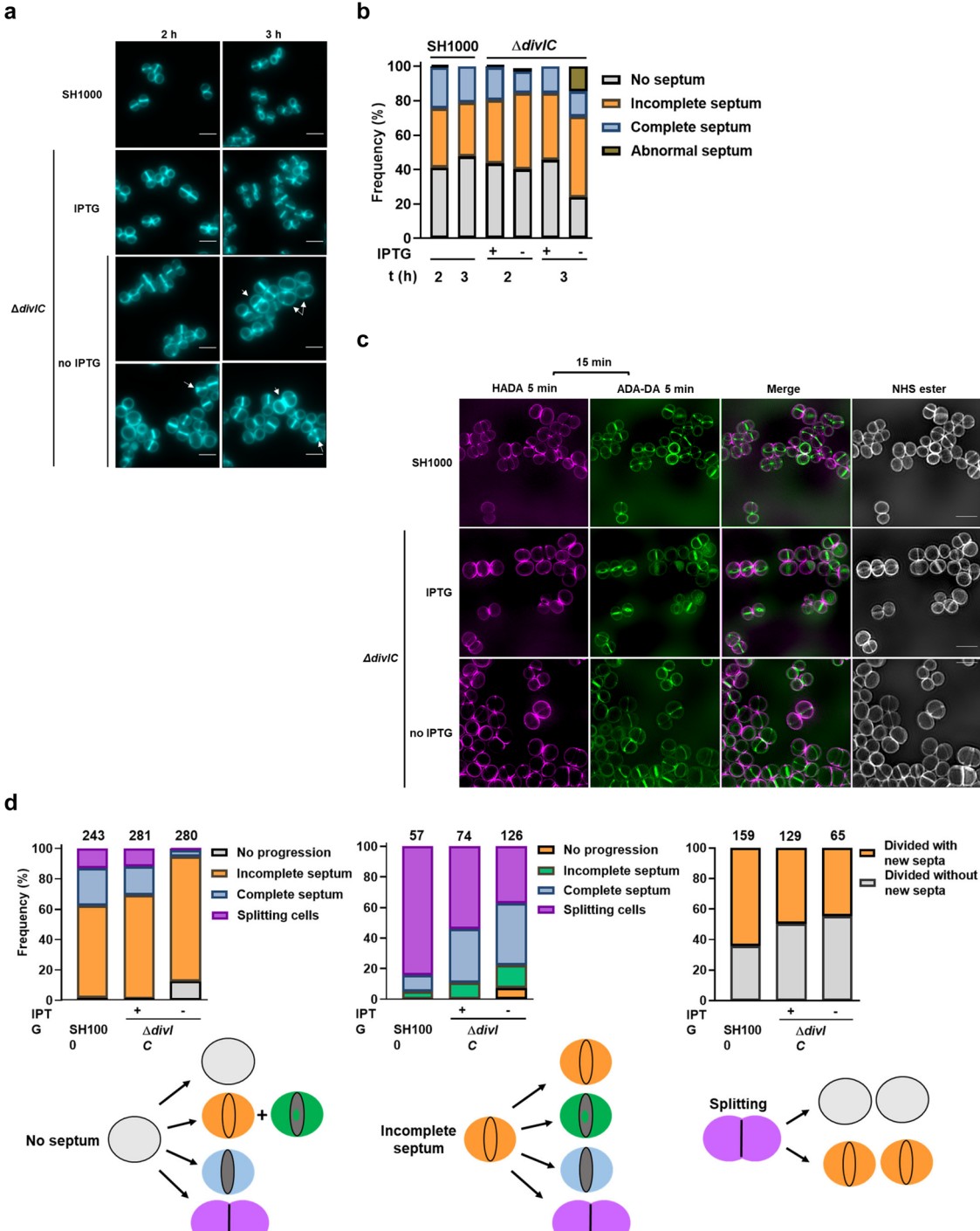

**Fig. 2 Loss of DivIC leads to defects in cell division. a** Fluorescence microscopy images of SH1000 and Δ*divIC* (SJF5450) cells grown for 2 and 3 h in the presence and absence of IPTG. PG was labelled by 30 min incubation with HADA. Arrows highlight cells with septal defects. **b** Quantification of cellular phenotypes based on HADA labelling (as shown in a), *n* = 300 cells per sample. **c** SIM microscopy images of SH1000 and Δ*divIC* grown in the presence or absence of IPTG. To follow cell cycle progression, after 2 h incubation with or without IPTG, HADA was added for 5 min, cells were washed and incubated for 15 min in fresh medium followed by the addition of ADA-DA for 5 min. Cells were subsequently stained with NHS-ester 555 to show cell morphology. **d** Cell cycle progression of SH1000 and Δ*divIC* cells grown with and without IPTG (from **c**). Cells with no septum (left panel), incomplete septum (middle panel), and splitting cells (right panel) were identified by HADA labelling and subsequent cell cycle progression was determined by ADA-DA incorporation patterns based on **c**. Total number of cells used for each analysis are shown on top of the columns (*n* = 459 (SH1000), 484 (Δ*divIC* plus IPTG), and 471 (Δ*divIC* no IPTG cells)). All images are average intensity projections of z stacks. Scale bars, 2 μm. Data is representative of two independent experiments.

For cells defined as splitting by HADA labelling, ~65% of SH1000 cells were found in the progression to a new round of division. Only a small difference was observed between cells grown with or without IPTG (50 and 45% respectively) (Fig. 2d right), suggesting that DivIC

is mainly involved in the ability of cells to transit through the intermediate stages of cell division (septum formation).

Transmission electron microscopy (TEM) imaging (Fig. 3a) revealed a significant increase in cell wall thickness of Δ*divIC* cells

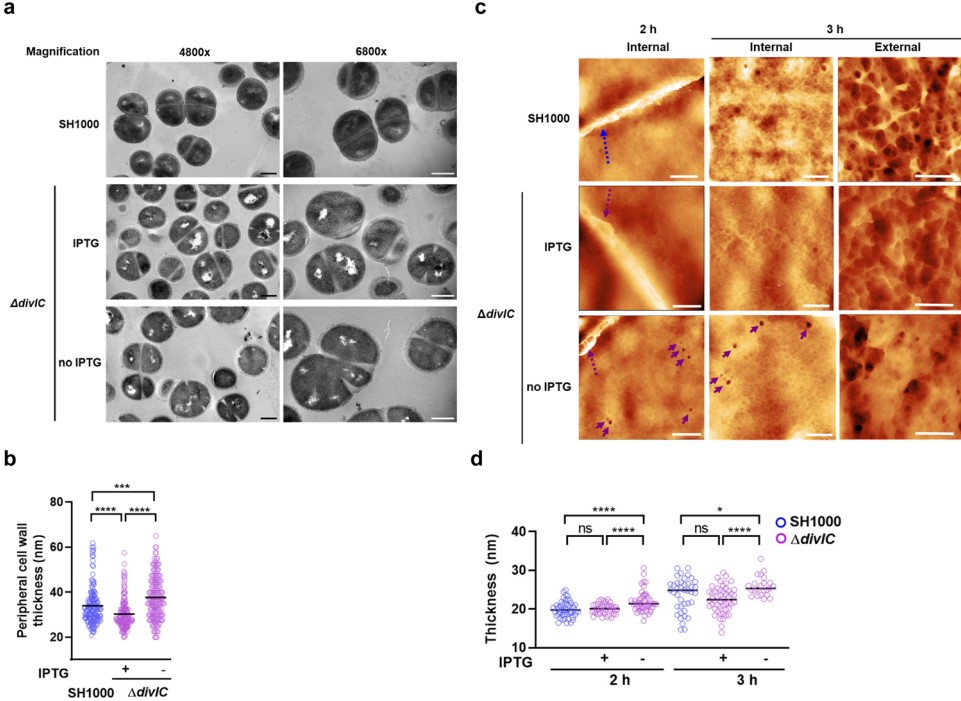

**Fig. 3 DivIC regulates the architecture of the cell wall. a** TEM of SH1000 and Δ*divIC* (SJF5450) grown for 2 h with or without IPTG. Scale bars 0.5 μm. **b** Measurements of peripheral cell wall thickness from TEM images. $n = 150$ cells per sample, $P$-values were determined by Mann–Whitney $U$-tests (****$P < 0.0001$; ***$P < 0.0005$). **c** AFM of the internal and external structures of cell walls from SH1000 and Δ*divIC* cells grown for 2 and 3 h in the presence and absence of IPTG. 2 h samples show piecrust development (dotted arrows) at the internal surface (scale bars, 100 nm, Z scales: 76 nm (SH1000), 73 nm (Δ*divIC* IPTG), 30 nm (Δ*divIC* no IPTG)). 3 h samples highlight internal (scale bars 50 nm, Z scales: 5.4 nm (SH1000), 6.3 nm (Δ*divIC* IPTG), 15 nm (Δ*divIC* no IPTG)) and external (scale bars 100 nm, Z scales: 40 nm (SH1000), 22 nm (Δ*divIC* IPTG), 60.5 nm (Δ*divIC* no IPTG)) architectural features (3 h). Arrows show deeper pores. **d** Cell thickness of single layer cell walls in air. Values were extracted from the AFM images using the 1D height distribution fitting method in Gwyddion. $n$ (left to right) = 50, 40, 48, 40, 50, 23. $P$-values were determined by unpaired two-tailed $t$-test with Welch's corrections (****$P < 0.0001$; *$P < 0.0120$).

grown for 2 h without IPTG compared to SH1000 or cultures incubated in the presence of the inducer ($P > 0.0001$) (Fig. 3b). The IPTG replete Δ*divIC* cells have thinner cell walls compared to SH1000 likely due to *divIC* expression levels from the non-native P$_{spac}$ promoter. After 3 h in the absence of IPTG, Δ*divIC* cells showed aberrant cell wall structures and lysis (Supplementary Fig. 3), suggesting the involvement of DivIC in PG metabolism.

To corroborate the effect of DivIC depletion on cell wall architecture, atomic force microscopy (AFM) of sacculi derived from Δ*divIC* grown without IPTG for 2 and 3 h was performed. Peripheral cell wall has a fine mesh at its internal face and a more open structure with large holes at its external[11]. During septation an internal band of PG, known as the piecrust, is first formed followed by the septal plate with the fine mesh architecture at its inner surface[9,11]. DivIC depleted cells show piecrusts and internal PG fine mesh where cells were observed to have deep pores ($12 \pm 2.7$ nm) compared to the wild type ($2.3 \pm 0.9$ nm) in both the inner-facing, peripheral (Fig. 3c and Supplementary Fig. 4) and septal cell wall (Supplementary Fig. 4). Lack of DivIC also led to an overall thickening of the peripheral wall ($P < 0.0001$; Fig. 3d and Supplementary Fig. 4).

**DivIC spatially regulates PG synthesis in cell division.** Localisation of PG synthesis at the subcellular level was analysed by measuring fluorescence at the cell periphery and septum, following the incorporation of HADA and ADA-DA in sequential 5 min pulses to identify cells which were actively septating (Fig. 4a). Cells of Δ*divIC* grown for 2 h in the absence of IPTG, that were undergoing septal synthesis, demonstrated a significant shift in fluorescence signal to the cell periphery compared to cells

grown with IPTG and SH1000 ($P < 0.0001$) (Fig. 4a, b). The IPTG replete Δ*divIC* cells have more peripheral synthesis compared to SH1000 likely due to *divIC* expression levels from the non-native P$_{spac}$ promoter. Importantly, this was observed using ADA-DA which is a marker of PG synthesis and not a PBP-associated exchange reaction[12].

In *S. aureus*, the bifunctional transglycosylase/transpeptidase, PBP2 is the primary PG transpeptidase catalysing the addition of new monomers into the PG envelope[15,36]. PBP2 is translocated from the cell periphery to the division site leading to the incorporation of PG at the septum during cell division[37]. Imaging a PBP2 fluorescent fusion (GFP-PBP2) expressed in the Δ*divIC* strain (Fig. 4c) showed a significant increase in the localisation of the protein at the periphery in dividing cells in the absence of IPTG compared to IPTG replete and SH1000 ($P < 0.0001$) (Fig. 4d).

A second PG transglycosylase, FtsW, and its cognate transpeptidase PBP1 (FtsW-PBP1) are involved in PG incorporation during cell division[38,39]. Compared to SH1000, microscopy images of Δ*divIC* cultures grown in the absence but not the presence of IPTG (Fig. 4e), also showed a significant increase in the peripheral localisation of an FtsW-GFP fusion compared to midcell ($P < 0.0001$) (Fig. 4f).

DivIC has been shown to interact, by bacterial two-hybrid analysis, with several divisome proteins including PBP2, FtsW, PBP1, DivIB and EzrA[40,41]. EzrA is an early division protein and, in contrast to the behaviour observed with PBP2 and FtsW, depletion of DivIC resulted in an increase in septally-located EzrA-GFP (Supplementary Fig. 5). Loss of DivIC had no effect on the localisation of GFP-DivIB (Supplementary Fig. 5).

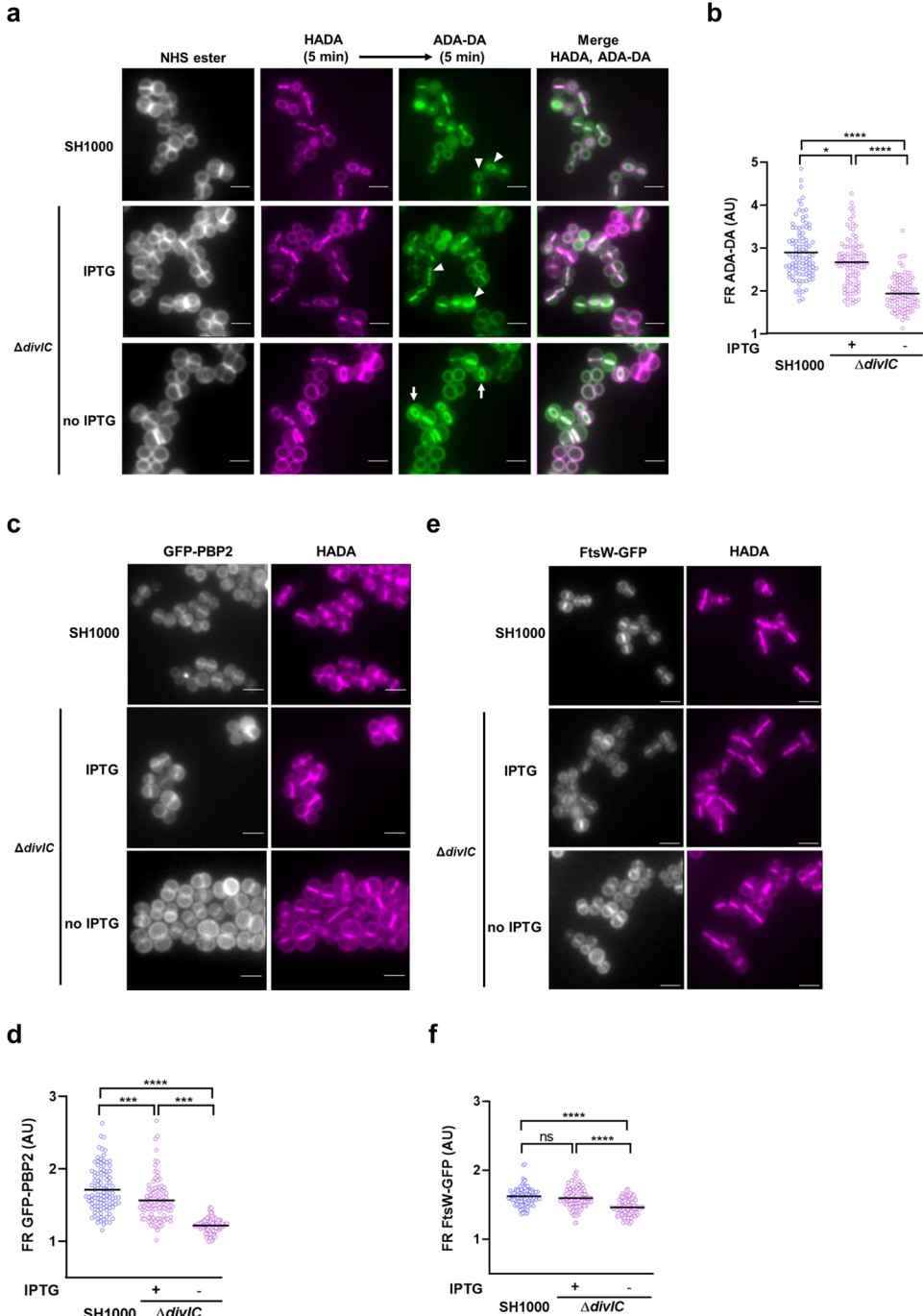

**Fig. 4 DivIC spatially regulates PBP-dependent PG synthesis during cell division. a** Fluorescence microscopy images of SH1000 and Δ*divIC* (SJF5450) cells grown for 2 h with and without IPTG and subsequently treated with HADA followed by ADA-DA (5 min each) to follow PG synthesis. Arrowheads show cells where PG synthesis is mainly observed at the septum. Arrows show exemplar cells with PG synthesis at the septum and cell periphery. Cells were counterstained with NHS-ester 555 to image cell walls. **b** Fluorescence ratios (FR) are the product of fluorescence signal at the septum divided by the signal at the cell periphery in cells with incomplete septa after incubation with ADA-DA. **c** Localisation of GFP-PBP2 in SH1000 (SJF5541) and Δ*divIC* (SJF5560) cells growing with and without IPTG for 2 h and incubated with HADA for 30 min to label PG. **d** FR between GFP-PBP2 fluorescent signal at the septum versus the cell periphery in cells with incomplete septa. **e** Localisation of FtsW-GFP in SH1000 (SJF5768) and Δ*divIC* (SJF5767) cells growing with and without IPTG for 2 h and incubated with HADA for 30 min to label PG. **f** FR between FtsW-GFP fluorescent signal at the septum versus the cell periphery in cells with incomplete septa. Each circle indicates a single cell, lines are mean of an *n* = 100. *P*-values were determined by Mann–Whitney *U*-tests (****P < 0.0001; ***P < 0.0002). All images are average intensity projections of z stacks. Scale bars, 2 µm. Data is representative of two independent experiments.

**The C-terminal domain of DivIC is essential for function, stability and septal localisation.** In *S. aureus*, DivIC has a predicted topology with a short N-terminal cytoplasmic domain, followed by a membrane spanning region and an exoplasmic, C-terminal domain of around 74 amino acids (Fig. 5a). Across the bacteria, DivIC amino acid conservation is concentrated in the extracellular domain (Fig. 5a, Supplementary Fig. 6) suggesting functional importance. Indeed, a role for DivIC and DivIB in the regulation of peptidoglycan synthesis at the septum has been proposed to rely solely on their external domains in *B. subtilis*[28]. To

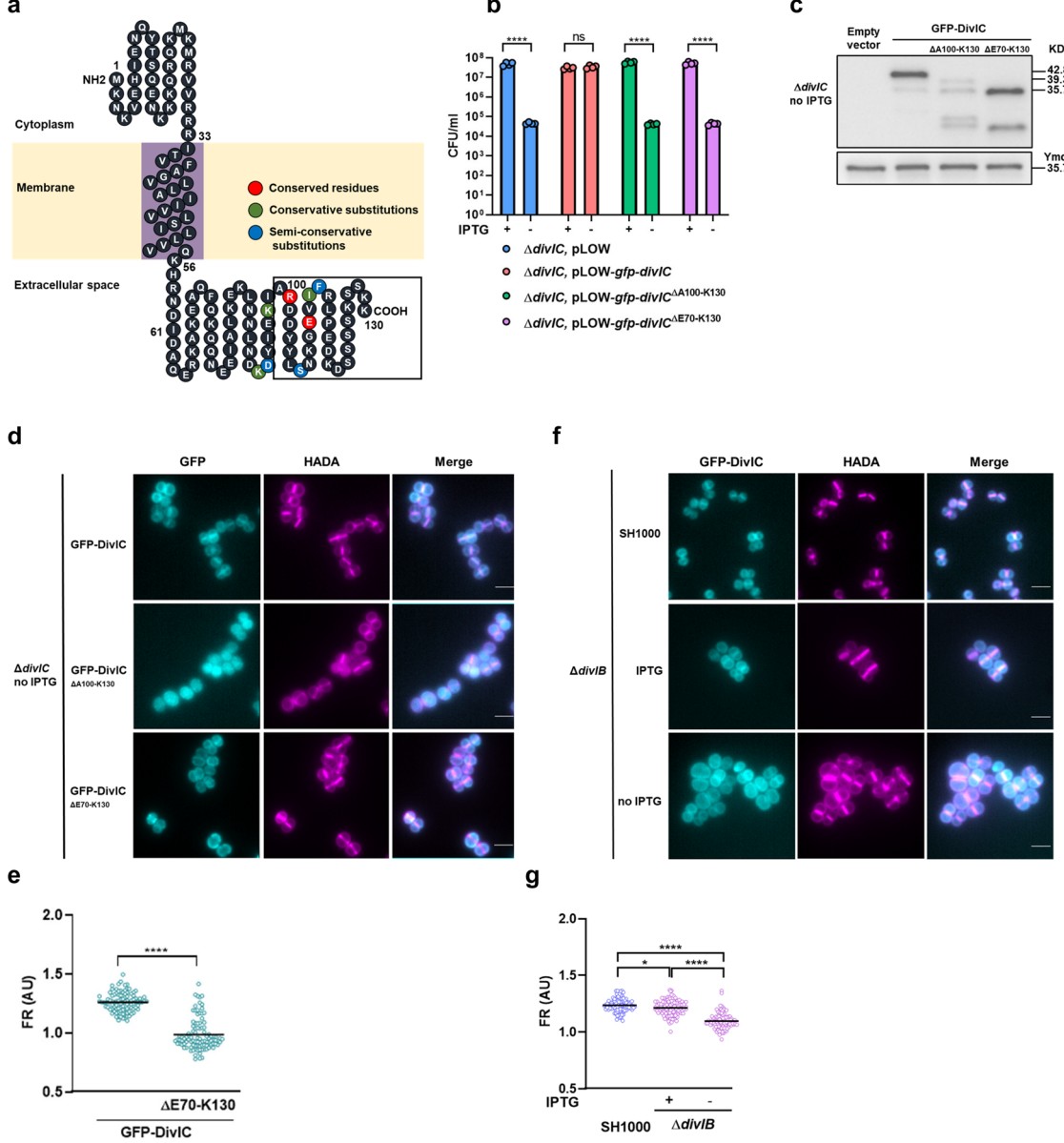

**Fig. 5 The C-terminal domain of DivIC is essential for function and septal localization. a** A hypothetical topological model of the *S. aureus* DivIC protein. **b** Viability test of Δ*divIC* carrying the empty vector (SJF5544) or expressing GFP-DivIC (SJF5473), GFP-DivIC$^{\Delta A100-K130}$ (SJF5553) and GFP-DivIC$^{\Delta E70-K130}$ (SJF5554) in the presence and absence of IPTG. Data represents the mean and standard deviation of at least three independent experiments. *P*-values were determined by two-tailed unpaired *t*-test (****$P < 0.0001$). **c** Western blot of whole cell lysates from the Δ*divIC* strain carrying the empty vector or expressing GFP-DivIC, GFP-DivIC$^{\Delta A100-K130}$, or GFP-DivIC$^{\Delta E70-K130}$. Cells were grown without IPTG for 2 h. Anti-GFP antibodies were used for detection. **d** Localisation of GFP-DivIC, GFP-DivIC$^{\Delta A100-K130}$ and GFP-DivIC$^{\Delta E70-K130}$ in Δ*divIC* cells grown without IPTG for 2 h and incubated with HADA for 30 min to label PG. **e** FR from GFP-DivIC, and GFP-DivIC$^{\Delta E70-K130}$ signals (from data based on **d**). **f** Localisation of GFP-DivIC in SH1000 (SJF5424) and Δ*divIB* (SJF5427) cells growing with and without IPTG for 2 h and incubated with HADA for 30 min to label PG. **g** FR from GFP-DivIC (from data based on **f**). **e** and **g** Each circle indicates a single cell, lines are mean of an $n = 100$. *P*-values were determined by Mann–Whitney *U*-tests (*$P = 0.0238$, ****$P < 0.0001$). All images are average intensity projections of z stacks of cells with incomplete septa. Scale bars, 2 μm. FR are the product of the fluorescence signal at the septum divided by the signal at the cell periphery in cells with incomplete septa. Data is representative of two independent experiments.

test the functional involvement of the exoplasmic domain of DivIC, we constitutively expressed native or C-terminally truncated DivIC (DivIC$^{\Delta A100-K130}$ and DivIC$^{\Delta E70-K130}$) in the Δ*divIC* strain. Only the complete protein restored the viability of Δ*divIC* cultures grown in the absence of IPTG, suggesting the final 31 amino acids of DivIC are essential for function (Supplementary Fig. 6).

The lack of function of the truncated DivIC protein could be related to loss of stability as a western blot of cell lysates showed no signal for DivIC$^{\Delta A100-K130}$ (Supplementary Fig. 6). However,

since the anti-DivIC antibodies were raised against the exoplasmic domain, we were unable to discern whether the lack of signal was due to protein degradation or the failure of the antibody to bind to truncated DivIC. To overcome this, fluorescent fusions of the complete (GFP-DivIC) and truncated DivIC (GFP-DivIC$^{\Delta A100-K130}$) were expressed in the Δ*divIC* background and antibodies raised against GFP used for protein detection. The complete form of DivIC but not the truncated version supported growth in the absence of IPTG (Fig. 5b). Lysates of Δ*divIC*

expressing GFP-DivIC$^{\Delta A100\text{-}K130}$ produced a much weaker signal than those expressing GFP-DivIC, supporting the need for the C-terminal tail to stabilise the protein (Fig. 5c). To confirm a role for the exoplasmic DivIC domain in cell division rather than simply protein stability a further GFP-DivIC truncation missing the entire C-terminal region (GFP-DivIC$^{\Delta E70\text{-}K130}$) was expressed in the $\Delta divIC$ background (Fig. 5b, c). This produced a strong signal with anti-GFP antibodies but did not support continued growth (Fig. 5b, c), demonstrating the essentiality of the DivIC exoplasmic domain. This also shows that it is loss of part of the exoplasmic domain that leads to DivIC protein stability rather than deletion of the entire domain.

GFP-DivIC localises to the cell periphery and mid-cell during cell division (Fig. 5d, e); however, GFP-DivIC$^{\Delta A100\text{-}K130}$ did not show a defined localisation and instead, fluorescence was seen evenly distributed within the cells, perhaps due to loss of stability as suggested by the immunoblot results (Fig. 5c, d). GFP-DivIC$^{\Delta E70\text{-}K130}$, localised at the cell periphery; however, unlike the complete protein, truncated DivIC was absent from the septum of cells that were actively dividing (Fig. 5d, e), indicating that the C-terminal domain is required for the localisation of DivIC at the division site. Localisation of DivIC at the septal site was also significantly reduced in the absence of DivIB, compared to IPTG replete and SH1000 (Fig. 5f, g), suggesting a functional interaction between DivIC and DivIB[40–42].

**DivIC interacts with cell wall through binding to teichoic acids**. *S. aureus* DivIB has been identified as a non-enzymatic peptidoglycan-binding protein, whose interaction with the cell wall depends on its extra cytoplasmic domain[24]. To explore the possibility that DivIC binds to components of the cell envelope, recombinant extracellular domain of DivIC (DivIC$_{ex}$) was expressed and purified (Supplementary Fig. 7) (amino acids 56 to 130, Fig. 5a). DivIC$_{ex}$ binds to purified cell walls from SH1000 and the interaction is significantly decreased when the preparation lacks wall teichoic acids (WTA), due to chemical treatment using HF (Fig. 6a and Supplementary Fig. 7), loss of WTA biosynthesis (*tarO*)[43] (Fig. 6b and Supplementary Fig. 7), or inability to ligate the polymer to PG (*lcpABC*)[44–46] (Supplementary Fig. 8). Cytochrome C, a protein of similar size and charge to DivIC, shows no WTA dependent binding to cell walls (Supplementary Fig. 7). By microscale thermophoresis analysis, we found that DivIC$_{ex}$ binds to soluble, muropeptide-free WTA isolated from SH1000 cell walls (Supplementary Fig. 9). Even though the binding is concentration dependent (Supplementary Fig. 9), no saturation was achieved, perhaps due to the heterogeneity and self-interaction of the WTA preparation at high concentrations.

The role of WTA in GFP-DivIC localisation was tested using $\Delta tarO$, revealing a significant enrichment of DivIC at the septum (higher FR values) in the absence of WTA (Fig. 6c, d). Also, depletion of DivIC did not affect the localisation of the WTA biosynthesis enzyme TarO, using a TarO-GFP fusion (Supplementary Fig. 10).

## Discussion
DivIB, DivIC and FtsL are recruited as late cell division proteins[23,26,30] with crucial roles in division. They form a trimeric complex in many organisms and have been shown to be mutually important for each other's stability. Indeed, in *B. subtilis* the structures of DivIC and FtsL are thermodynamically unstable, likely requiring protein-protein interactions for stabilisation[47]. In *E. coli*, FtsB protects FtsL against RasP cleavage[48], and the lack of FtsB leads to degradation of FtsL[49]. FtsBQL may function as a unit directly interacting with PBP1b and FtsW-PBP3 to inhibit their enzymatic activity in *E. coli*[50]. In *B. subtilis*, FtsL is required

to stabilise DivIC but not DivIB and impairs the localisation of DivIC, DivIB and PBP2B to the division site[26,30]. Degradation of DivIC in the absence of FtsL depended on the presence of DivIB, which has consequently been suggested as a negative regulator of DivIC[23]. In *S. aureus*, DivIB and DivIC functions seem to be independent. Indeed, loss of either of these proteins does not lead to mutual instability or loss of other division proteins. In fact, in DivIB or DivIC depleted cells there was a significant increase in FtsL levels over time (Supplementary Fig. 2). Further support for independent protein function is provided by the DivIB requirement for DivIC recruitment to the septum but not vice versa. Loss of DivIC also results in increased levels of EzrA at the septum, an early cell division protein[40,41], consistent with an orchestrated sequence of divisome component recruitment.

Cell division requires a carefully choreographed succession of events from initiation, through septal development to scission, thus ensuring maintenance of cellular integrity and fidelity across the generations. We have found DivIC in *S. aureus* to be essential for continued growth and division and its depletion to lead to cell death. Death of the cells is associated with an increase in cell volume and the appearance of holes at the inner surface of the cell wall, strikingly in the incomplete septum. Wall spanning holes have recently been found to appear as a consequence of inhibition of PG synthesis by cell wall antibiotics, such as β-lactams and vancomycin, leading to lack of ability to maintain turgor[16].

Loss of DivIC leads to a delay in cell division associated with increased PG synthesis at the cell periphery to the detriment of septal development, an observation supported by the demonstration of thicker peripheral cell walls in these cells. The major, essential PG synthetase in *S. aureus*, responsible for bulk septal synthesis is PBP2[15,51]. It's preferential recruitment to the septum is dependent on several factors including apparent removal of receptive synthesis sites around the periphery[52], the presence of PG transpeptidase substrates[37], and its interaction with many divisome proteins including DivIB, DivIC and FtsL[40]. The direct interaction between DivIC and PBP2 may be crucial for septal PBP2 localisation as DivIC depletion leads to a redistribution to the periphery. DivIC is therefore essential for appropriate septum formation during division. We have proposed that septal plate formation occurs after the production of the PG piecrust that forms the foundations for the septum. This constitutes a cell division checkpoint for the recruitment of those components required for bulk septum synthesis. DivIB is required to get through this checkpoint as its loss leads to cells stuck at the piecrust to septal plate transition stage[24]. After this PG architectural transition, septal plate formation itself is not a simple process, as in *S. aureus* and other Gram positive organisms a concentric ring architecture is first formed at its core to be exposed on the surface after scission[11,53]. Underneath the rings and forming the bulk of the septum is a finer mesh with an architecture similar to the peripheral wall, which it becomes after scission[9,11]. We hypothesise that PBP1 maybe responsible for the central ring architecture of PG, and PBP2 of the bulk fine mesh of the septum, as it is around the cell periphery. FtsW acts with PBP1 as a cognate pair and both are required for division[38,39]. FtsW also shows a decrease in septal localisation in the DivIC depleted cells and so, together with the loss of PBP2 recruitment, this will lead to the slowing of septal plate synthesis with such dramatic consequences for the cell.

If DivIC is coordinating the temporal and spatial control of septal synthesis then how is this manifested? DivIC interacts with several members of the divisome as well as DivIB and FtsL[40]. This can be carried out via its membrane spanning and/or exoplasmic domain. We found the exoplasmic domain of DivIC to not only be essential for cell viability but also for localisation of the protein at the septum. As DivIB has previously been shown to be a PG binding protein[24], the cell wall binding capabilities of DivIC were

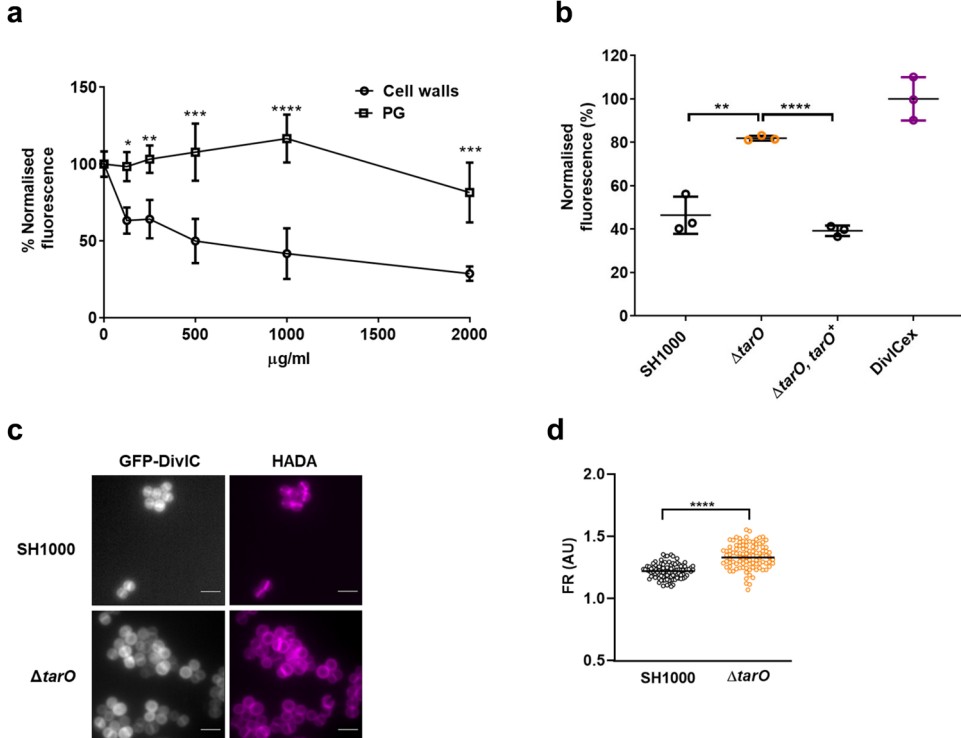

**Fig. 6 DivIC interacts with the cell wall in a teichoic acid-dependent manner. a** Fluorescence of Cy2-labelled DivIC$_{ex}$ measured after 5 min incubation with purified cell walls or PG from SH1000. Cy2-labelled DivIC$_{ex}$ fluorescence was determined in the supernatants after centrifugation to determine the levels of cell wall- or PG-unbound protein. Mean and standard deviations of three independent experiments are shown. *P*-values were determined by two-tailed *t*-test corrected for multiple comparison using the Holm–Šídák method. **b** Cell walls (0.25 mg/ml) from SH1000, Δ*tarO* (SJF5289) mutant or complemented Δ*tarO* (SJF5396) were incubated with Cyt2-labelled DivIC (500 nM) for 5 min and fluorescence measured in the supernatant after centrifugation. Mean and standard deviations of three independent repeats are shown. *P*-values were calculated by two-tailed *t*-test. **c** Localisation of IPTG-induced GFP-DivIC in SH1000 (SJF5398) and Δ*tarO* (SJF5395) cells grown to mid exponential phase and incubated with HADA for 30 min to label PG. Scale bars, 2 μm. **d** FR between GFP-DivIC fluorescence signal at the septum versus the cell periphery. Each circle indicates a single cell, lines are mean of an *n* = 100. *P*-values were determined by Mann–Whitney *U*-tests. All images are average intensity projections of z stacks of cells with incomplete septa. Data is representative of two independent experiments. (\**P* < 0.05, \*\**P* < 0.01, \*\*\**P* < 0.005, \*\*\*\**P* < 0.0001).

investigated. This revealed an affinity of DivIC for cell walls containing WTA and for purified WTA. Interestingly, loss of TarO, and thus WTA synthesis, led to increased levels of DivIC at the septum, indicating DivIC's ability to also bind septally located proteins or other components.

The binding of the essential cell division proteins DivIB and DivIC to cell wall components provides a hypothetical framework for their roles in septum synthesis coordination. For the cell to be able to divide, it must not only make, but also recognise, the dynamic structure of the wall during the cell cycle. This begs the question as to what are the cell wall, chemical and architectural checkpoints that form the cues for ensuing progression? The *S. aureus* cell envelope consists of a membrane, with embedded LTA, and a thick PG layer that affords mechanical strength. Covalently bound to the PG are WTA that alters the charge and has a number of cellular functions[43]. Collectively the teichoic acids are essential for cell viability[43] and have been proposed to be responsible for the maintenance of the exoplasm (the region between the cytoplasmic membrane and PG[54]), in which the final stages of PG biosynthesis takes place[55]. Not only are there PG architectural changes during division but also teichoic acid dynamics. Even though the biosynthetic apparatus for both LTA and WTA biosynthesis has been shown to be septally located[56,57], it has been proposed that these polymers are absent from the septum[58,59]. WTA promotes a high degree of PG cross-linking by regulating the migration of the PG synthase PBP4 to the division site[54]. It was speculated that immature forms of WTA present exclusively at the septum act as spatial-temporal signals for

PBP4 recruitment to the division site[57]. There is also evidence that WTA protects PG from enzymatic cleavage by exogenous and endogenous hydrolases[58,60]. Thus, WTA may regulate both cross-linking and degradation of PG during cell division. Whether the septum contains immature WTA forms, supported by the presence of TarO[57], a lack of WTA[58], or differential display associated with individual PG architectures, remains a fundamental question and may allude to its role in cell division.

Here we have characterised DivIC, with a pivotal role in cell division, forming a regulatory link between architectural development of the septum and the temporal and spatial requirement for those components responsible for its synthesis. As there is conservation of division components across divergent species, elucidation of underlying mechanisms in one organism has important implications for understanding across bacteria.

## Methods

**Bacterial strains and growth conditions**. A complete list of strains, plasmids and oligonucleotides used in this work are listed in Supplementary Table 1, 2 and 3 respectively. All strains were cultured at 37 °C with aeration. *E. coli* strains were cultured in Luria-Bertani (LB) broth or agar plus 100 μg/ml ampicillin. *S. aureus* strains were grown in tryptic soy broth (TSB) (Bacto, BD 211825) or agar (TSA) and, when required, added with 5 μg/ml erythromycin plus 25 μg/ml lincomycin (Ery), 10 μg/ml chloramphenicol (Cm), 5 μg/ml tetracycline (Tet), and 50 μg/ml kanamycin (Kan).

**Construction of strains and plasmids**. *E. coli* NEB5a was used for construction of all vectors according to formerly described methods[61,62]. Plasmids were introduced

into restriction-deficient *S. aureus* strain RN4220 by electroporation and moved into *S. aureus* SH1000 and its derivatives by phage transduction as previously described[63] unless otherwise stated. All chromosomal modifications and plasmid constructs were verified by sequencing (Source Bio International).

**SJF5450 (*divIC* conditional lethal strain (Δ*divIC*)).** For the construct of pMAD-U-*tet*-D*divIC*, using SH1000 genomic DNA as a template, 904 bp upstream (U*divIC*) and 960 bp downstream of *divIC* (D*divIC*) were amplified. U*divIC* amplification was amplified with primers MT55 and AK22 including BglII and NotI restriction sites at the 5' and 3' sites respectively. For D*divIC*, primers AK23 and MT56 including NotI and EcoRI restrictions sites at the 5' and 3' sites respectively were used. Subsequently, both U*divIC* and D*divIC* PCR products were digested with NotI and ligated. The ligation mixture was used as a template for PCR amplification with oligos MT55 and MT56 rendering UdivIC-NotI-DdivIC. Both the PCR product and the pMAD vector were digested with BglII and EcoRI and ligated rendering pMAD-U-D*divIC*. For cloning of the tetracycline cassette into the NotI site, oligos MT57 and MT58 containing NotI restriction sites were used for the amplification of the *tet* cassette from pOB. pMAD-U-D*divIC* and the amplified *tet* cassette were digested with NotI and ligated producing pMAD-U-*tet*-D*divIC*. Steps for the chromosomal integration of pMAD-U-*tet*-D*divIC* by single crossover recombination and subsequent excision of the pMAD derivative by double crossover were performed as described previously[64]. Finally, pGL485 carrying a constitutively expressed copy of *lacI* was introduced by transduction to provide a tight control of the P*spac*-divIC system.

**SJF5395 (Δ*tarO*, pCQ11-P*spac*-gfp-divIC*).** *S. aureus* lacking teichoic acids (Δ*tarO*) is naturally resistant to Φ11 infection[65]. Therefore, phage transduction is not possible. To overcome this, the pCQ11~P*spac*-gfp-divIC construct was transformed into RN4220, the plasmid was extracted by conventional miniprep from transformant cells treated with lysostaphin[66] for 1 h at 37 °C and used for the transformation of SH1000 Δ*tarO* (SJF5289).

**pALB26.** The *divIC* extracellular domain encoding K56 to K130 (DivIC$_{ex}$) was amplified from SH1000 using oligos ALB21Fw and ALB22Rv. The amplicon and vector (pET21d) were cut with NcoI and XhoI and ligated for the expression of DivIC$_{ex}$-6xHis under control of P*lac*.

**pLOW-*divIC*, pLOW-*gfp-divIC*, pLOW-*divIC*$^{ΔE70-K130}$, pLOW-*gfp-divIC*$^{ΔE70-K130}$, pLOW-*divIC*$^{ΔA100-K130}$ and pLOW-gfp-divIC$^{ΔA100-K130}$.** For pLOW-*divIC*, pCQ11-*gfp-divIC* was used as a template for the amplification of the whole plasmid lacking the *gfp* gene with oligos MT80Fw and MT81Rv and the PCR product was religated using Q5® Site-Directed Mutagenesis Kit according to the manufacturer instructions to obtain pCQ-*divIC*. pCQ-*divIC* and pCQ11-*gfp-divIC* were then used as templates for the amplification of P*pcn*-divIC and P*pcn*-gfp-divIC respectively with oligos MT87Fw and MT88Rv, oligos MT87Fw and MT90Rv were used for the amplification of P*pcn*-divIC$^{ΔE70-K130}$, P*pcn*-gfp-divIC$^{ΔE70-K130}$, and MT87Fw and MT89Rv for the amplification of P*pcn*-divIC$^{ΔA100-K130}$ and Ppcn-gfp-divIC$^{ΔA100-K130}$. The PCR products were cloned into pLOW cut with BamHI and AflII according to NEBuilder® HiFi DNA Assembly Cloning Kit (E5520S) instructions.

**DivIC and DivIB depletion for phenotypic and growth analysis.** SJF5450 (Δ*divIC*) and SJF3883 (Δ*divIB*) strains were cultured in TSB containing Cm and 500 μM IPTG to exponential phase (OD$_{600nm}$ 0.4). Cells were washed twice with TSB by centrifugation and used to inoculate fresh medium to an OD$_{600nm}$ 0.05 in the presence of Cm with or without 500 μM IPTG. For growth, changes in OD$_{600nm}$ were monitored every hour for 5 h, while for phenotypic studies (microscopy), samples were taken after 2 and 3 h incubation.

**Viability tests (plating efficiency).** Samples were cultured in TSB containing Cm, Ery and 500 μM IPTG to exponential phase (OD$_{600nm}$ 0.4). Cell were washed twice with TSB by centrifugation and the pellets were resuspended in 1 ml TSB and diluted in TSB to an OD$_{600nm}$ of 0.6. Decimal serial dilutions were prepared and plated in TSA containing Cm and Ery with or without 1 mM IPTG and plates were incubated for 24 h. Relative plating efficiency (%CFU) was expressed as the number of CFU/ml from plates without IPTG to CFU/ml from those with IPTG multiplied by 100%.

**Whole cell lysates and cell fractionation.** For preparation of whole cell lysates of SJF5450 (Δ*divIC*) and SJF3883 (Δ*divIB*), cell cultures were grown according to the procedure described for "phenotypic and growth analysis" (above), and samples were taken after 1, 2 and 3 h incubation in the presence and absence of IPTG. For whole cell lysates and fractionation of SJF5289 (Δ*tarO*) cultures were grown to exponential phase without antibiotics (OD$_{600nm}$ 0.8). Cells were harvested by centrifugation, washed twice, and resuspended in 1 ml ice-cold TBSI (50 mM Tris-HCl, pH 7.0 containing protease inhibitors (Sigma, S8830)). Cell suspensions were added to lysing matrix tubes (0.1 mm glass beads, Merk, BMSD113101) and broken by repeated cycles of lysis using an MP Biomedicals FastPrep 24 Homogeniser. Supernatant (whole cell lysate) was separated from glass beads by centrifugation at

low speed (5 min, 2500 x g) at 4 °C. For separation of cell wall, cytoplasmic and membrane fractions, whole cell lysates were further centrifuged for 15 min at 22,000 x g to separate supernatant from unbroken cell walls and debris. Insoluble pellet (cell walls) and supernatant (cytoplasmic content plus membranes) were further separated by centrifugation at 26,000 x g for 10 min and cell walls washed once, resuspended in 500 μl TBSI and stored at -20 °C. For separation of cytoplasm and membrane, supernatant was ultracentrifuged at 250,000 x g for 1 h at 4 °C and the soluble fraction (cytoplasm) separated and stored at -20 °C. Pellet (membranes) were washed once by repeating the ultracentrifugation cycle and resuspended in 200 μl TBSI. Protein concentration was determined by Pierce™ BCA protein Assay Kit (Thermo Scientific, 23227) and samples normalized for Western Blot.

**Antibodies.** Anti-FtsZ, anti-DivIB[24] and anti-DivIC polyclonal antibodies were obtained from rabbits immunised with purified his-tagged recombinant *S. aureus* FtsZ, DivIB and DivIC (BioServ). Anti-PBP1 and anti-PBP2 antibodies were produced from rabbits immunized with sPBP1A-BAP and His-tagged PBP2 (Eurogentec, Belgium), and purified as previously described[67]. Anti-FtsL polyclonal antibodies were generated from a rabbit immunized with small synthetic peptides (MAVEKVYQPYDEQVYC and CNDNVKVVRSNGEAKN) followed by affinity antigen specific IgG purification (Eurogentec). Anti-GFP antibodies were from Merck (SAB4701015).

**Western Blot.** A total of 10 μg protein per sample were separated by SDS-PAGE (12% w/v) followed by transfer to activated PVDF membranes. Membranes were blocked in 5% (w/v) skimmed-milk dissolved in TBST (20 mM Tris-HCl, pH 7.6, 17 mM NaCl, 0.1% (v/v) Tween-20) and incubated with polyclonal primary antibodies (1:1000 dilution) for 12 h at 4 °C. For detection, goat anti-rabbit IgG conjugated to horseradish peroxidase (1:10,000 dilution, BioRad, 1706515) and Clarity Western ECL Substrate (BioRad, 1705061) were used according to the manufacturers' instructions. Syngene G:BOX Chemi XX9 was used for chemiluminescent detection.

**Expression and purification of DivIC$_{ex}$.** SJF3165 strain was cultured at mid exponential phase and supplemented with 1 mM IPTG followed by a further 6 h incubation. Cells were harvested, resuspended in START buffer (NaPO4 100 mM, NaCl 500 mM) and lysed by 3 cycles of -80 °C freeze-thaw followed by sonication. The soluble fraction containing the recombinant protein was loaded onto a 5 ml His tag protein purification column (HisTrap™, 17524802) previously loaded with NiSO$_4$ and equilibrated with START buffer. Protein was eluted with START buffer containing imidazole and dialysed using a Spectra/Por 3 dialysis membrane (3.5 KDa MWCO) against PBS for 3 h at 4 °C. Protein concentration was determined using a Micro BCA™ Protein Assay Kit (Thermo, 23235) and stored at -20 °C.

**Cy2 and Alexa Fluor 647 labelling of DivIC$_{ex}$.** For Cy2 labelling, equal amounts (1 mg) of DivIC$_{ex}$ and Fluorolink Cy2™ Reactive Dye (Amersham, 23031) were mixed in PBS and incubated in the dark for 5 h at room temperature with gentle rotation.

For Alexa Fluor 647 labelling, DivIC$_{ex}$ (0.5 mg) was mixed with 100 μl of 2 mM Alexa Fluor™ 647 NHS Succinimidyl Ester (AF647) (Thermo) and 5 μl of triethylamine and the mix incubated in the dark for 24 h at room temperature with gentle rotation. The tube was then left open for 1 h for evaporation of triethylamine.

In both cases, the protein-dye conjugates were dialysed using a Spectra/Por 3 dialysis membrane (3.5 KDa MWCO) against PBS containing 10% BSA (w/v) at 4 °C to trap any unbound dye. Dialysis cycles were repeated four times.

**Isolation of cell walls and purification of peptidoglycan.** Extraction was performed as previously described[68]. Bacterial cells were grown to stationary phase, centrifuged, and resuspended in Tris-HCl, pH 7.5 containing 2% SDS (w/v). Samples were boiled for 10 min to kill bacteria, then centrifuged, washed, and resuspended in 1 ml water. Suspensions were added to lysing matrix tubes containing 0.1 mm silica beads (Lysing Matrix B, MP Biomedicals, 116911050-CF) and cells were broken by repeated cycles of lysis using an MP Biomedicals FastPrep 24 Homogeniser. Broken cells (supernatant) were separated from the lysing matrix by centrifugation at low speed (5 min, 2400 x *g*) at 4 °C. Sacculi (pellet) were harvested from the supernatant by centrifugation at 220,000 x *g* for 15 min and resuspended in 4% (w/v) SDS followed by boiling for 30 min. After centrifugation, pellet was resuspended in 50 mM Tris-HCl, pH 7.5 containing 3% SDS (w/v), 1.25 mM EDTA and 50 mM DTT and boiled for 30 min to remove any non-covalently bound proteins and remaining lipid-linked molecules. SDS was removed by repeated washes with water and resuspended in 50 mM Tris-HCl, pH 7.5 containing pronase (2 mg/ml) and the suspension incubated at 60 °C for 90 min to degrade cell wall-binding proteins. Pure cell walls were resuspended in water and stored at -20 °C or used for the purification of peptidoglycan.

Peptidoglycan was produced by stripping cell walls of cell wall polymers, such as WTA, by incubation in 48% (v/v) HF for 48 h at 4 °C. Purified peptidoglycan was then washed with alternating buffer (Tris 50 mM, pH 7.5) and H$_2$O until pH 5 was reached. Pure peptidoglycan was resuspended and stored at -20 °C.

**Quantification of cell walls and peptidoglycan.** Cell walls or peptidoglycan pellets were quantified by freeze drying in a pre-weighed tube for 12 h (ScanVac Cool Safe 55-4 Pro 3800). Tubes were re-weighed and stock concentrations were calculated in mg/ml.

**Binding of DivIC$_{ex}$ to cell walls and PG (pulldown assay).** Purified DivIC$_{ex}$ (100 μg/ml) unlabelled or labelled with Cy2 was incubated with increasing concentrations of cell wall or peptidoglycan in binding buffer (20 mM sodium citrate, 10 mM MgCl$_2$, 0.1% Tween (v/v), 10 μg/ml BSA, pH 5) for 5 min at room temperature. Samples were centrifuged for 10 min at 16,000 x $g$, and soluble and insoluble fractions were separated. For mixtures containing unlabelled DivIC$_{ex}$, both soluble and insoluble fractions were analysis by SDS-PAGE. For Cy2-labelled protein, fluorescence was measured in soluble fractions to quantify the levels of unbound Cy2-DivICex using a Victor2TM, Wallac plate reader.

**Wall teichoic acids (WTA) purification.** Isolated cells walls were incubated with 0.1 M NaOH for 72 h at room temperature with mild rotation followed by centrifugation at 22,000 x $g$ for 10 min to separate the supernatant (containing WTA) from the insoluble material. The pH was neutralised using acetic acid 5% (v/v) and supernatant was then dialysed using 0.5-1.0 KDa tubing (SpectrumTM labs) against water. Dialysis was repeated four times to minimise the presence of salt in the sample. The dialysed sample was transferred into a pre-weighed tube, frozen at -80 °C and lyophilized (ScanVac Cool Safe 55-4 Pro 3800) for 12 h. Finally, tubes were re-weighed and WTA yield calculated in mg/ml.

**Microscale thermophoresis analysis (MST).** MST was performed on a Monolith NT.115 (Nanotemper) using nanotemper control software. Fluor 647-labelled DivIC$_{ex}$ (100 nM) (using Alexa Fluor 647 Invitrogen™ A37573) was added to increasing concentrations of purified *S. aureus* WTA in binding buffer (20 mM sodium citrate, 10 mM MgCl$_2$, 0.1% Tween (v/v), 10 μg/ml BSA, pH 5). Samples were loaded on Monolith NT.115 Premium capillaries (Nanotemper). Measurements were carried out using the red detector at 10% excitation and medium MST power at 21 °C. Quality controls such as sample aggregation, ligand autofluorescence and starting fluorescent levels were performed using the inbuilt software. The binding check protocol was used for comparison of fluorescent levels of protein alone (negative control) and protein-ligand complexes.

**Labelling *S. aureus* cells with D-amino acids (DAAs).** For labelling of nascent peptidoglycan, SH1000 and SJF5450 (Δ*divIC*) were cultured in TSB containing Cm and 500 μM IPTG to exponential phase (OD$_{600nm}$ 0.4). Cells were washed twice with TSB by centrifugation and used to inoculate fresh medium to an OD$_{600nm}$ 0.05 in the presence of Cm with or without 500 μM IPTG. After 2 h incubation, 1 mM HADA (Bio-Techne, 6647) was added and cultures were incubated for a further 5 min. Cells were then washed twice with TSB at 37 °C by centrifugation and resuspended in fresh medium with or without IPTG. To follow cell cycle progression, cells were incubated for 15 min prior to the addition of 500 μM ADA-DA for 5 min. For PG synthesis analysis, ADA-DA was added immediately after washes. Cells were then washed twice and resuspended in PBS at 4 °C. Cells were then labelled with NHS-ester 555 (Invitrogen, A20009) in PBS at 8 μg/ml for 15 min, fixed and labelled with Fluor 488 Alkyne (Merk, 761621) (for fluorescent labelling of the ADA-DA azide group) by the click reaction (copper(I)-catalysed alkyne-azide cycloaddition) using the Click-iT™ Cell Reaction Buffer Kit (Invitrogene, C10269) according to the manufacturer instructions.

**Cell fixation for optical microscopy.** All samples were treated with 2% (w/v) paraformaldehyde (PFA) for 30 min at room temperature and PFA removed by washing cells with water prior to imaging.

**Widefield epifluorescence microscopy.** Fixed cells mounted onto poly-L-Lysine coated slides using Slow Fade Diamond mounting medium (Thermo Fisher, S36967) and imaged using a Nikon Ti inverted microscope fitted with a Lumencor Spectra X light engine. Images were obtained with a 100x PlanApo (1.4 NA) oil objective 1.518 RI oil and an Andora Zyla sCMOS camara was used for detection.

**OMX microscopy (Structured Illumination Microscopy (SIM)).** Coverslips (High-precision, 1.5H, 22 ± 22 mm, 170 ± 5 mm, Marienfeld, 0107052) were immersed in 1 M KOH, sonicated for 15 min, and washed prior to incubation in poly-L-Lysine solution for 30 min. Coverslips were then washed and dried with pressurized air. Fixed cells resuspended in water were dried onto the coverslips with nitrogen and mounted on slides using Slow Fade Diamond mounting medium (Thermo Fisher, S36967). SIM was performed using a v4 DeltaVision OMX 3D-SIM system fitted with a Blaze module (Applied Precision, GE Healthcare, Issaquah, USA) using laser illumination. Images were taken in five phase shifts and three angles for each slice with Z-steps of 0.125 nm. Softworx software (GE Healthcare) was used for reconstruction with OTFs optimisation for 1.516 immersion oil.

**Expression of fluorescent fusions.** To assess localization of GFP-PBP2, FtsW-GFP, GFP-DivIC, GFP-DivIC$^{ΔE70-K130}$, GFP-DivIC$^{ΔE70-K130}$, EzrA-GFP, GFP-DivIB and TarO-GFP in SH1000 and the Δ*divIC* strain, cultures were grown in TSB containing Cm, Ery and 500 μM IPTG to exponential phase (OD$_{600nm}$ 0.4). Cell were washed twice with TSB by centrifugation and used to inoculate fresh medium to an OD$_{600nm}$ 0.05 in the presence of antibiotics with or without 500 μM IPTG. After 1.5 h incubation, 1 mM HADA was added, and cultures incubated for a further 30 min. Cells were harvested by centrifugation and resuspended in PBS followed by fixation. Δ*tarO* was cultured in the presence of Kan and 1 mM IPTG for the expression of GFP-DivIC. At mid exponential phase (OD$_{600nm}$ 0.4), 1 mM HADA was added and cultures incubated for a further 30 min before harvesting cells for fixation.

**Cell volume measurements.** Cell volume was calculated as previously described[13]. The software Fiji was used for measurements of the long and short axis of cells and the volume was calculated according to a prolate spheroid shape with volume:

$$V = \frac{4}{3}\pi ab^2$$

$V$ is the volume, $a$ and $b$ are the radii for the long and short axis, respectively.

**Calculation of fluorescent ratios (FR).** Fluorescence intensity was measured using Image J/Fiji software. Fluorescence ratios (FR) were calculated as the fluorescence intensity at the septum (only cells with incomplete septum were considered and measurements taken between the cell periphery and the annulus) divided by the fluorescence at the lateral cell wall (i.e., average of fluorescence at opposite poles of the wall taken perpendicularly to the septum). FR for 100 cell per strain were plotted (modified from[57]). SH1000 cells stained with Nile Red for 5 min to stain the cytoplasmic membrane were fixed and microscopy images were used to calculate the FR of peripheral versus mid-cell membranes (Supplementary Fig. 5). The average FR for stained membranes was ~1.3. Thus, any FR values above 1.3 were considered as a measurement of increased septal localisation. Values below 1.3 were regarded as a measurement of enriched peripheral localisation (modified from[57]).

**Transmission electron microscopy and cell wall thickness measurements.** Cell preparation for electron microscopy was performed as described previously[68].

Cell pellets (5 ml cultures) were fixed overnight at 4 °C in 2.5% (w/v) glutaraldehyde. Samples were washed with PBS and resuspended in 2% (w/v) aqueous osmium tetroxide (Merk, 75632) for secondary fixation (2 h at room temperature). Cells were washed with PBS and dehydrated by incubating with increasing concentrations of ethanol (75% v/v, 95% v/v and 100% (v/v) ethanol) 15 min each. Ethanol was removed and samples were incubated with propylene oxide (Merck, 540021) to complete dehydration. Samples were mixed with a 1:1 mix of propylene oxide and Epon resin (TAAB Laboratories, E208) and incubated overnight at room temperature to allow infiltration. Resin was removed, and the excess of propylene oxide evaporated at room temperature. Two consecutive incubations of the samples with pure Epon resin (4 h each) were performed and cells were embedded in fresh resin. Resin polymerisation was achieved by incubation at 60 °C for 48 h. Thin sections (80 nm) were produced using an Ultracut E Ultramicrotome (Reichert-Jung) and mounted onto 200-square mesh copper TEM grids (Merck, G4776-1VL) pre-treated with Pyroxylin (1.5%) (w/v, in amyl acetate) film. Sections were stained in aqueous uranyl acetate (3%) (w/v) for 30 min, washed with dH$_2$O and stained with Reynold's lead citrate for 5 min. Citrate was removed by washing with dH$_2$O. A FEI Tecnai T12 Spirit Transmission Electron Microscope operating at 80 kV was used for imaging. A Gatan Orius SC1000B bottom mounted CCD camera recorded the images.

Measurements were performed using Image J/Fiji software. Cell wall thickness values are the average of four measurements around the cell periphery avoiding septal areas.

**Atomic force microscopy.** Sacculi stocks (non-HF treated) from all samples were diluted to ~10 x volume with HPLC grade water. The resulting suspension was incubated for 60 min (optimal time was determined empirically for each batch of sample, using 10 μl of sacculi dilution) on a Cell-Tak coated mica disc[11]. Samples were washed 3 times with HPLC grade water to ensure good adhesion, the sacculi were then dried onto the substrate with flowing nitrogen and rehydrated again for imaging in 150 mM KCl, 10 mM Tris, pH 7.8. AFM images were collected using a Bruker (Santa Barbara) Dimension FastScan AFM. FastScan-D probes (Bruker) with a nominal spring constant of 0.25 N m$^{-1}$ and nominal resonant frequency (in liquid) of 110 kHz were used for all images. However, for the thickness measurements images were taken in air using SCOUT probes (NuNano) with a nominal spring constant of 42 N m$^{-1}$ and a nominal resonant frequency (in air) of 350 kHz. All AFM data in liquid was acquired with "PeakForce Tapping" mode with a typical amplitude of 100–150 nm, frequency of 2 kHz and setpoint force of 1–3 nN on a Bruker Dimensions FastScan equipment. All AFM data in air was acquired with "Intermittent Tapping" mode with a typical free amplitude of 10–12 nm and setpoint amplitude of 7–9 nm on a Bruker Dimensions FastScan equipment.

The thickness measurements performed on images taken in air were performed with standard Gwyddion 1D height distribution method (selecting an area containing background and a single leaflet of peptidoglycan). The number of cells analysed for the 2 h and 3 h samples varied from $n = 40$–50 (for SH1000 and samples containing IPTG), and $n = 23$–48 (for the samples without IPTG). Therefore, the conclusions and comparison extracted from this data is statistically relevant. However, the depth measurements of the pores (marked with arrows) were measured from $n = 2$ cells (for the samples without IPTG) and $n = 1$ (for SH1000).

**DivIC topological model**. The hypothetical topological model of DivIC was obtained using Gapped BLAST and PSI-BLAST[69].

**Protein alignment**. Clustal Omega[70] was used for the DivIC protein sequences alignment.

**Statistics and reproducibility**. GraphPad Prism version 9.0 was used to perform statistical analysis. Statistical tests, sample sizes and biological repeats are described in figure legends. Mann–Whitney $U$-tests were used to determine differences in cell volumes and fluorescent rates derived from fluorescence microscopy and SIM images, and to determine differences in cell thickness obtained from TEM images. For AFM images, $P$-values were determined by unpaired two-tailed $t$-test with Welch's corrections. Two-tailed unpaired $t$-test were applied to compare cell viability data and western blot signals. For pull-down assays, the $t$-test was corrected for multiple comparison using the Holm-Šídák method. A one-way ANOVA with Dunnett's multiple comparison was used for in vitro binding tests to cell walls and WTA.

**Reporting summary**. Further information on research design is available in the Nature Research Reporting Summary linked to this article.

## Data availability

The data that support the findings of this study, including the numerical data behind the graphs and the microscopy images used for analysis are available in the Online Research Data (ORDA) figshare from the University of Sheffield, https://doi.org/10.15131/shef.data.c.6240234.

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

## Acknowledgements

This research was funded in part by the Wellcome Trust (212197/Z/19/Z). For the purpose of open access, the authors have applied a CC BY public copyright licence to any Author Accepted Manuscript version arising from this submission. This work was also funded by the Medical Research Council (MR/S014934/1) and UKRI Strategic Priorities Fund (EP/T002778/1). Imaging work was performed at the Wolfson Light Microscopy Facility funded by the Wellcome Trust (WT093134AIA) and the MRC SHIMA award (MR/K015753/1). We are grateful to Kasia Wacnik for help and advice.

## Author contributions

M.T.-T., O.C, A.F.K., A.H., J.K.H. and S.J.F. designed research; M.T.-T., O.C., A.F.K., L.P.-L., and L.L. performed research; M.T.-T., O.C., A.F.K., L.P.-L., J.K.H. and S.J.F. analyzed data; and M.T.-T. and S.J.F. wrote the manuscript.

## Competing interests

The authors declare no competing interests.
