## [Peer Review File · Communications Biology]

Reviewers' comments:

Reviewer #1 (Remarks to the Author):

In their manuscript, Tinajero-Trejo and colleagues characterize the role of DivIC in the *S. aureus* division process. They generate a DivIC depletion strain and use it to show that DivIC regulates septal peptidoglycan synthesis through its interaction with the peptidoglycan synthetase PBP2. Specifically, they show that the extracellular domain of DivIC is responsible for its function and localization to the septum during division, and that this domain interacts directly with the cell wall via wall teichoic acids. The authors use clear writing and convincing experiments to demonstrate the essential role that DivIC plays in regulating cell wall synthesis during division. Overall, the data are rigorous, quantitative, and clearly presented, and support most of the stated conclusions. This work sheds light on mechanisms of cell division in *S. aureus* and should be of interest to those in the bacterial cell biology community. However, while the authors provide compelling data to show that DivIC is important for cell wall synthesis during division, the focus on PBP2 (and exclusion of FtsW-PBP1) as a target of its regulation seems too narrow to rule out alternative/additional functions for DivIC.

Specific Points:

1. The authors examine the role of DivIC in the regulation of cell wall biosynthesis by examining its effects on PBP2. However, other cell wall synthetic enzymes play a primary role in septum synthesis, notably FtsW and PBP1 (Reichmann et al 2019). These enzymes have been shown to be the primary PG synthetic enzymes involved in cell division in a variety of bacteria, however there is only brief mention of them in the discussion. In *E. coli* and other Gram-negative species, the ortholog of DivIC (FtsB) is part of the FtsQLB complex that directly interacts with FtsW and FtsI to regulate cell wall biosynthesis (e.g. Marmont and Bernhardt 2020 PNAS, Liu et al 2015 Mol Micro). As the authors note, conservation of these factors across bacterial phyla suggests a common mechanism of action/regulation and the possibility that DivIC regulates FtsW-PBP1 seems valid. Could DivIC be interacting with FtsW and FtsI to regulate PG synthesis during division?
2. The authors use the Δ divIC strain grown in the presence of IPTG (DivIC expressed) as a control for their experiments. However, several of these controls differ significantly from their SH1000 (WT) strain (Figs. 1e, 2d, 3b, 4b, 4d, and 5g) in a variety of the experiments they analyzed. The authors do not comment on these differences or provide an explanation for these observations. Are these differences significant enough to affect the conclusions they draw from experiments performed in the absence of IPTG? Furthermore, the authors sometimes compare their strains grown without IPTG to their SH1000 strain and other times to the +IPTG control strain. It would be preferred to compare to both the replete and WT controls throughout to account for both strain background and DivIC levels.
3. The authors claim that "DivIC and DivIB play independent roles in cell division in *S. aureus*" (Lines 114-115). Can the authors clarify/explain this claim? Do you mean independent from each other? This conclusion does not seem supported by the data provided.
4. In Line 63 and Line 165 different spellings of piecrust were used (pie-crust vs. piecrust).
5. Line 82-84: Additional obligate intracellular bacteria lack orthologs of one or more of the FtsQ/DivIB, FtsL, FtsB/DivIC complex (Otten et al 2019, Mol Micro).
6. Fig. 3b, d. Why are box and whisker plots used here where scatter plots are used for all other similar types of data? Showing all data points (i.e. in scatter plots) is most informative.
7. Figure 3a: Are the images on the right higher magnification than on the left? Some indication of the difference between the two columns could be helpful for clarity.

8. Supplementary Figure 1a: No IPTG or -IPTG label should be added to bottom image for clarity.

Reviewer #2 (Remarks to the Author):

Summary:

In this manuscript by Tinajero-Trejo & Carnell et al the function(s) of the *S. aureus* cell division protein DivIC is investigated. They show that DivIC is an essential member of the PG-synthesis family, by directly influencing the septal recruitment of PBP2, the major PG-synthase in *S. aureus*.

The manuscript had obviously been crafted (and re-crafted) extensively, as it is very well and clearly written. The experimental flow is logical and mostly easy to follow. I believe the authors have done a fantastic job.

The authors show snapshots of deformation of Δ divIC cells over time with a membrane stain. Something that comes to mind for the future, it would be interesting to see the dynamics of a fluorescently labelled septa (e.g., using FP fusions to division proteins and/or FDAs and high speed time-lapse imaging) in these cells.

Overall, this study furthers our understanding of a fundamental biological process, the division of a cell. I support publication of this manuscript.

There are a few outstanding issues that the authors may want to address.

Major concerns:

- The WB and the quantified levels of FtsL are not convincing (supple fig 1), this is of importance since as there is a claim that DivIC and DivIB are not required for division stability. Could this be rectified?

Minor concerns:

- Why was the imaging stopped at 3 h, could this be extended to 5h, as the growth was followed this long.

- Missing delta symbols in figure 1c top right graph. And divIB should be italic.

- Formatting of the figures is not optimal.

- While mathematically the P values are sound, is there a more convincing way of presenting the differences in the fluorescence differences data?

For example in Figure 5g the p has one * (SH1000 vs Δ divIB(+)).

This is a quite general question, not necessarily something I expect you to fix here.

Replies to Comments

For ease, changes to the revised version of the manuscript are highlighted in red.

Reviewers' comments:

Reviewer #1 (Remarks to the Author):

*In their manuscript, Tinajero-Trejo and colleagues characterize the role of DivIC in the *S. aureus* division process. They generate a DivIC depletion strain and use it to show that DivIC regulates septal peptidoglycan synthesis through its interaction with the peptidoglycan synthetase PBP2. Specifically, they show that the extracellular domain of DivIC is responsible for its function and localization to the septum during division, and that this domain interacts directly with the cell wall via wall teichoic acids. The authors use clear writing and convincing experiments to demonstrate the essential role that DivIC plays in regulating cell wall synthesis during division. Overall, the data are rigorous, quantitative, and clearly presented, and support most of the stated conclusions. This work sheds light on mechanisms of cell division in *S. aureus* and should be of interest to those in the bacterial cell biology community. However, while the authors provide compelling data to show that DivIC is important for cell wall synthesis during division, the focus on PBP2 (and exclusion of FtsW-PBP1) as a target of its regulation seems too narrow to rule out alternative/additional functions for DivIC.*

See reply to Specific Point 1 below concerning FtsW-PBP1.

Specific Points:

1. The authors examine the role of DivIC in the regulation of cell wall biosynthesis by examining its effects on PBP2. However, other cell wall synthetic enzymes play a primary role in septum synthesis, notably FtsW and PBP1 (Reichmann et al 2019). These enzymes have been shown to be the primary PG synthetic enzymes involved in cell division in a variety of bacteria, however there is only brief mention of them in the discussion. In *E. coli* and other Gram-negative species, the ortholog of DivIC (FtsB) is part of the FtsQLB complex that directly interacts with FtsW and FtsI to regulate cell wall biosynthesis (e.g. Marmont and Bernhardt 2020 PNAS, Liu et al 2015 Mol Micro). As the authors note, conservation of these factors across bacterial phyla suggests a common mechanism of action/regulation and the possibility that DivIC regulates FtsW-PBP1 seems valid. Could DivIC be interacting with FtsW and FtsI to regulate PG synthesis during division?

This is an important question and so we have created an FtsW-GFP fusion. In the absence of DivIC, there is an increase in peripheral FtsW compared to septal. This data has been included in Figure 4 (panels e and f) and explanatory text added to the Results and Discussion sections (l. 196-203, 319-321).

2. The authors use the Δ divIC strain grown in the presence of IPTG (DivIC expressed) as a control for their experiments. However, several of these controls

differ significantly from their SH1000 (WT) strain (Figs. 1e, 2d, 3b, 4b, 4d, and 5g) in a variety of the experiments they analyzed. The authors do not comment on these differences or provide an explanation for these observations. Are these differences significant enough to affect the conclusions they draw from experiments performed in the absence of IPTG? Furthermore, the authors sometimes compare their strains grown without IPTG to their SH1000 strain and other times to the +IPTG control strain. It would be preferred to compare to both the replete and WT controls throughout to account for both strain background and DivIC levels.

The levels of production of DivIC in the presence of IPTG in the Pspac constructs are unlikely to be the same as SH1000 (WT), which could account for the observed phenotypes. We have highlighted this within the manuscript (l. 106-109). The important comparison is between WT and no IPTG (DivIC depletion). We have highlighted the DivIC replete situation and have made the comparisons explicit throughout the manuscript.

For Fig. 1e l. 112-114

For Fig. 2d l. 156-161

For Fig. 3b l. 165-166

For Fig. 4b, d l. 187-188, 198-203

3. The authors claim that “DivIC and DivIB play independent roles in cell division in *S. aureus*” (Lines 114-115). Can the authors clarify/explain this claim? Do you mean independent from each other? This conclusion does not seem supported by the data provided.

This is an important point. Both DivIB and DivIC are required for cell division overall but their depletion leads to different phenotypes. Thus, their roles are linked but are not entirely co-dependent. We have modified the text to clarify this issue (l. 119, 281-288).

4. In Line 63 and Line 165 different spellings of piecrust were used (pie-crust vs. piecrust).

Corrected

5. Line 82-84: Additional obligate intracellular bacteria lack orthologs of one or more of the FtsQ/DivIB, FtsL, FtsB/DivIC complex (Otten et al 2019, Mol Micro).

This is now highlighted in the introduction and the citation added (l. 83-85).

6. Fig. 3b, d. Why are box and whisker plots used here where scatter plots are used for all other similar types of data? Showing all data points (i.e. in scatter plots) is most informative.

Scatter plots are now used throughout.

7. Figure 3a: Are the images on the right higher magnification than on the left? Some indication of the difference between the two columns could be helpful for clarity.

The right hand images are at higher magnification. Labels have been added to the top of the columns for clarification (Fig. 3a).

8. Supplementary Figure 1a: No IPTG or -IPTG label should be added to bottom image for clarity.

Corrected (Fig. S1a).

Reviewer #2 (Remarks to the Author):

Summary:

In this manuscript by Tinajero-Trejo & Carnell et al is the function(s) of the *S. aureus* cell division protein DivIC investigated. They show that DivIC is an essential member of the PG-synthesis family, by directly influencing the septal recruitment of PBP2, the major PG-synthase in *S. aureus*.

The manuscript had obviously been crafted (and re-crafted) extensively, as it is very well and clearly written. The experimental flow is logic and mostly easy to follow. I believe the authors have done a fantastic job.

The authors show snapshots of deformation of Δ divIC cells over time with a membrane stain. Something that comes to mind for the future, it would be interesting to see the dynamics of a fluorescently labelled septa (e.g., using FP fusions to divisome proteins and/or FDAAs and high speed time-lapse imaging) in these cells.

This is a great idea for future studies as suggested.

Overall, this study furthers our understanding of a fundamental biological process, the division of a cell. I support publication of this manuscript.

There are a few outstanding issues that the authors may want to address.

Major concerns:

- The WB and the quantified levels of FtsL are not convincing (supple fig 1), this is of importance since as there is a claim the DivIC and DivIB are not required for divisome stability. Could this be rectified?

We have repeated the Western Blots using a new antibody preparation to give a clearer, signal. Three independent biological repeats were done and significance calculated (Fig. S2).

Minor concerns:

- Why was the imaging stopped at 3 h, could this be extended to 5h, as the growth was followed this long.

We have done labelling at 4 and 5 hr but by this time a significant proportion of the cells are dead, with evidence of collapse or crumples. Thus, we are not confident to carry out volume measurements on this population. An example of NHS-ester labelled cells at 5 h is shown below (scale bar 2 μ m), with arrows highlighting the deformed cells. We would prefer not to include these in our analysis as the morphology is likely *post mortem*.

- Missing delta symbols in figure 1c top right graph. And divIB should be italic.

The labels have been corrected (Fig. 1c).

- Formatting of the figures is not optimal.

We have checked the formatting throughout.

- While mathematically the P values are sound, is there a more convincing way of presenting the differences in the fluorescence differences data? For example in Figure 5g the p has one * (SH1000 vs Δ divIB(+)). This is a quite general question, not necessarily something I expect you to fix here.

We agree that using the actual p values is more exact but often this is difficult to do on a small figure panel without loss of clarity. The * is commonly used.

EVIEWERS' COMMENTS:

Reviewer #1 (Remarks to the Author):

In their manuscript Tinajero-Trejo and colleagues characterize the role of DivIC in the *S. aureus* division process. After review, they have now included data showing that in the absence of DivIC there is an increase in the peripheral localization of FtsW, a highly conserved PG synthase involved in division, thereby strengthening their claim that DivIC regulates PG synthesis during division. They have also addressed our concerns regarding differences in their Δ divIC controls compared to their SH1000 (WT) strain by providing explanations for these differences throughout the manuscript. Minor labeling and clarity issues have also been addressed and corrected. The authors use clear and rigorous data to support their conclusion that DivIC plays an essential role in regulating cell wall synthesis during division, with only a couple of minor issues that remain.

1. In the abstract in lines 29-32 they claim that DivIC regulates PG synthesis by influencing the recruitment of PBP2 to the site of division. In their manuscript they also show that DivIC effects recruitment of FtsW to the site of division (Fig. 4e and 4f). We suggest that FtsW be included in their abstract along with PBP2 as the major PG synthases that DivIC regulates during division.

2. For clarity purposes, in figures 4b, 4d, and 4f it would be helpful to include labels (either on the axis or above the graph) of what is being fluorescently labeled for each FR analysis (e.g. HADA, GFP-PBP2, FtsW-GFP).

3. In lines 89-90 "...however how it, and it cell division partners..." should be corrected to "...however how it, and its cell division partners..."

Reviewer #2 (Remarks to the Author):

The authors have done a great job! I have no further comments.

REVIEWERS' COMMENTS:

Reviewer #1 (Remarks to the Author):

In their manuscript Tinajero-Trejo and colleagues characterize the role of DivIC in the *S. aureus* division process. After review, they have now included data showing that in the absence of DivIC there is an increase in the peripheral localization of FtsW, a highly conserved PG synthase involved in division, thereby strengthening their claim that DivIC regulates PG synthesis during division. They have also addressed our concerns regarding differences in their Δ divIC controls compared to their SH1000 (WT) strain by providing explanations for these differences throughout the manuscript. Minor labeling and clarity issues have also been addressed and corrected. The authors use clear and rigorous data to support their conclusion that DivIC plays an essential role in regulating cell wall synthesis during division, with only a couple of minor issues that remain.

1. In the abstract in lines 29-32 they claim that DivIC regulates PG synthesis by influencing the recruitment of PBP2 to the site of division. In their manuscript they also show that DivIC effects recruitment of FtsW to the site of division (Fig. 4e and 4f). We suggest that FtsW be included in their abstract along with PBP2 as the major PG synthases that DivIC regulates during division.

The abstract has been modified to include reference to FtsW (lines 31-32)

2. For clarity purposes, in figures 4b, 4d, and 4f it would be helpful to include labels (either on the axis or above the graph) of what is being fluorescently labeled for each FR analysis (e.g. HADA, GFP-PBP2, FtsW-GFP).

Labels have been included

3. In lines 89-90 "...however how it, and it cell division partners..." should be corrected to "...however how it, and its cell division partners..."

This has been corrected as suggested

Reviewer #2 (Remarks to the Author):

The authors have done a great job! I have no further comments.